# Tooth Malformations, DMFT Index, Speech Impairment and Oral Habits in Patients with Fetal Alcohol Syndrome

**DOI:** 10.3390/ijerph16224401

**Published:** 2019-11-11

**Authors:** Moritz Blanck-Lubarsch, Dieter Dirksen, Reinhold Feldmann, Cristina Sauerland, Ariane Hohoff

**Affiliations:** 1Department of Orthodontics, University Hospital Münster, Albert-Schweitzer-Campus 1, 48149 Münster, Germany; hohoffa@uni-muenster.de; 2Department of Prosthodontics and Biomaterials, University Hospital Münster, Albert-Schweitzer-Campus 1, 48149 Münster, Germany; dieter.dirksen@ukmuenster.de; 3Department of Pediatrics, University Hospital Münster, Albert-Schweitzer-Campus 1, 48149 Münster, Germany; feldrei@uni-muenster.de; 4Institute of Biostatistics and Clinical Research, University of Münster, Schmeddingstraße 56, 48149 Münster, Germany; christina.sauerland@ukmuenster.de

**Keywords:** fetal alcohol spectrum disorder (FASD), fetal alcohol syndrome (FAS), tooth malformations, oral habits

## Abstract

*Background:* Fetal alcohol spectrum disorder (FASD) is a developmental disorder with severe negative lifetime consequences. Although knowledge about the harmfulness of alcohol consumption during pregnancy has spread, the prevalence of fetal alcohol spectrum disorder is very high. Our study aims at identifying fetal alcohol syndrome (FAS)-associated dental anomalies or habits, which need early attention. *Methods:* Sixty children (30 FAS; 30 controls) were examined prospectively. Swallowing pattern, oral habits, breastfeeding, speech therapy, ergotherapy, physiotherapy, exfoliation of teeth, DMFT (decayed, missing, filled teeth) index, modified DDE (developmental defects of enamel) index and otitis media were recorded. *Results:* Swallowing pattern, exfoliation of teeth, and otitis media were not significantly different. Significant differences could be found concerning mouthbreathing (*p* = 0.007), oral habits (*p* = 0.047), age at termination of habits (*p* = 0.009), speech treatment (*p* = 0.002), ergotherapy, physiotherapy, and breastfeeding (*p* ≤ 0.001). DMFT (*p* ≤ 0.001) and modified DDE (*p* = 0.001) index showed significantly higher values for children with fetal alcohol syndrome. *Conclusions:* Children with fetal alcohol syndrome have a higher need for early developmental promotion such as speech treatment, ergotherapy, and physiotherapy. Mouthbreathing, habits, and lack of breastfeeding may result in orthodontic treatment needs. High DMFT and modified DDE indexes hint at a higher treatment and prevention need in dentistry.

## 1. Introduction

Fetal alcohol spectrum disorder (FASD), which is a completely avoidable developmental disorder with resulting negative lifetime consequences receives far too little attention and is underestimated even by specialists across the world [1,2]. The cause of FASD is alcohol consumption during pregnancy, and although knowledge about the harmfulness of alcohol during pregnancy has spread, there still is an estimated worldwide prevalence of 7.7 per 1000, with regional differences [3]. According to Popova et al., the prevalence in certain sub-populations, such as children in care, in special education, in specialized clinics or Aboriginal populations is even 10 to 40 times higher [4]. May et al. found a prevalence of 1.1% to 5.0% in first-grade schoolchildren in four US communities [5].

The diagnosis FASD is subdivided into four subgroups according to the severity of the symptoms: The fetal alcohol syndrome (FAS) is the most severe form, followed by the partial fetal alcohol syndrome (pFAS), alcohol-related birth defects (ARBD), and alcohol-related neurodevelopmental disorder (ARND) in decreasing order [6]. A major problem with FASD is the complexity of diagnosis. According to the four-digit code by Astley and Clarren four diagnostic criteria are crucial for the verification of FASD: (1) Growth deficiency, (2) facial phenotype, (3) damage or dysfunction of the central nervous system (CNS), and (4) gestational exposure to alcohol [7]. For the fourth point (gestational exposure to alcohol), it is obvious that not every mother would respond truthfully to this question and for children in foster care this question might be impossible to answer [8]. For number two, facial phenotype, the fading of abnormal facial features with age can complicate diagnosis [8,9]. The facial phenotype is diagnosed using three components: Short palpebral fissure length, smooth philtrum, and thin upper lip. Diagnosis of smooth philtrum and thin upper lip can be performed using the Lip-philtrum guide by Astley and Clarren, which consists of photographs to be visually compared to the patient during diagnosis [10].

Recent studies by Blanck-Lubarsch et al. found metrically measurable, significant differences in philtrum depths, eye and nose parameters in children with FAS [11,12,13]. In addition, malocclusion in the sense of a significantly higher prevalence of crossbites in children with FAS was described [14]. According to a study by Muggli et al., even low levels of alcohol can influence craniofacial development and the timing of alcohol exposure might influence which areas are affected [15]. A systematic review by Hendricks et al. found that children with FASD show a language delay of up to three years of age [16]. Terband et al. described differences in speech development in boys with FASD compared to healthy controls and concluded that speech impairment resulted from a combination of deficits in multiple subsystems [17]. Anomalies of enamel formation can result from genetic, systemic, local or unknown causes [18]. Animal studies could show that alcohol exposure during pregnancy influences the secretory function in the ameloblasts, which in turn influences enamel formation [19]. An animal study by Sant’Anna et al. could show that alcohol exposure results in reduced development of the tooth germ and has the most severe effect on enamel matrix formation [20]. The extent of enamel defects can be assessed by the modified DDE (Developmental Defects of Enamel) index as defined by Clarkson et al. [21]. An example of enamel defects in one of our study patients can be seen in Figure 1a.

A study by Luz et al. found that short durations (<6 months) of breastfeeding resulted in a significantly higher prevalence of nonnutritive sucking habits, which in turn are associated with higher prevalence of class 2 malocclusions (dental distoocclusion, mandibular retrognathism, maxillary prognathism, or combinations) [22]. Breastfeeding for 12 or more months seems to be beneficial for dental occlusion [22,23,24]. Oral habits, such as thumb sucking or using pacifiers, may result in the formation of class 2 relationships, open bites, or crossbites [25,26,27]. Mouthbreathing can result in malocclusion with increased or reduced overjet (sagittal distance between the upper and lower incisors), anterior and posterior crossbite (one or more teeth in the upper arch has or have a more lingual position than the corresponding antagonist tooth in the lower arch), open bite (Figure 1b), and displacement of contact points [28]. An example of an open bite in one of our study patients can be seen in Figure 1b.

Our study aims at identifying FAS associated dental anomalies or habits, which need early attention and treatment.

## 2. Materials and Methods 

### 2.1. Study Design, Setting and Participants

A total of 60 children in the mixed dentition (mean age = 8.5 years), 30 children with FAS (15 female, 15 male) and 30 controls (12 female and 18 male) were examined in this prospective study. The children with FAS were recruited by a specialist in the Pediatric Department of the University Hospital Muenster, who introduced our study to the children with verified FAS diagnosis. The controls were voluntary children from local schools. The examination was performed by one single specialist in the Orthodontic Department of the University Hospital Muenster. Inclusion criteria were mixed dentition for both groups, verified FAS diagnosis for the FAS group and absence of disorders, syndromes or diseases with dento- or craniofacial characteristics for the controls. Exclusion criteria were primary or permanent dentition as well as completed or current orthodontic treatment.

### 2.2. Variables and Data Sources/Management

For this study, a standardized orthodontic examination protocol was performed for all children and exfoliation of teeth (normal, premature or delayed), swallowing pattern (adult/infantile), mouthbreathing, the modified DDE (developmental defects of enamel) and the DMFT (Decayed, Missing, Filled Teeth) index were recorded [21,29,30,31,32,33].

For the assessment of enamel anomalies, the modified DDE index for epidemiological studies as described by Clarkson et al. was used [21]. This index consists of grading into code 0 to 9 for the type of defect for every single permanent tooth:

Normal: 0

Demarcated opacities: White/cream: 1Yellow/brown: 2

Diffuse opacities: Diffuse-lines: 3Diffuse-patchy: 4Diffuse-confluent: 5Confluent/patchy + staining + loss of enamel: 6

Hypoplasia
Pits: 7Missing enamel: 8

Any other defects: 9

The DMFT (decayed, missing, filled teeth) index is recommended by the WHO [30] and was first described by Klein and Palmer in 1938 [31]. It is used to assess the amount of decayed, missing, and filled teeth. The maximum score was set at 28, the minimum score was 0.

Ratios were calculated for modified DDE and DMFT index, since in the mixed dentition the number of permanent and deciduous teeth can vary between individuals. The modified DDE index was divided by the number of permanent teeth, whereas the DMFT index was divided by permanent and deciduous teeth. In addition, the number of teeth with enamel defects was divided by the number of permanent teeth.

All children and their legal guardians were questioned concerning the following aspects: Oral habits, age at stopping the habit, speech therapy, ergotherapy, physiotherapy, breastfeeding, duration of breastfeeding, and otitis media.

### 2.3. Bias

To minimize bias, control children were not recruited from an orthodontic university department but from local schools to avoid selection of extreme malocclusions and oral phenotypes that might have an influence on facial contours. All examinations and measurements were performed by the same experienced orthodontist. All data were blinded regarding study groups prior to measurements and statistical evaluation.

### 2.4. Statistical Analysis

All analyses were performed with the software IBM^®^ SPSS^®^ Statistics 25 (IBM, Armonk, NY, USA). Data were described by absolute frequencies for categorical variables. Continuous variables were characterized by the arithmetic mean (M), standard deviation (SD), median (MD) and range (minimum, maximum) and visualized via boxplots depicting median (horizontal line), interquartile range (box), data range (whiskers) with dots representing outliers or via bar chart. The pre-analysis of the data showed that the null-hypothesis of normally distributed data could be rejected by Kolmogorov–Smirnov tests in most subgroups and the skewness of the data was extreme in most subgroups. To apply a consistent analysis strategy throughout the dataset we chose a non-parametric analysis strategy for uniform comparisons in all cases. Mann–Whitney *U*-test and Fisher’s exact test were used to assess differences between FAS and control groups. All analyses were regarded as explorative and *p*-values interpreted descriptively. Therefore, no adjustment for multiple testing was performed. The local two-sided significance level was set at *p* < 0.05. 

### 2.5. Ethical Approval

The study was approved by the ethics committee of the medical association of Westphalia–Lippe and the Department of Medicine, University of Muenster, Germany, study-code 2012-196-f-S. The investigation was performed in compliance with the current revision of the Declaration of Helsinki, and with the International Conference on Harmonisation Good Clinical Practice (ICH-GCP) guidelines. Written informed consent for performing the clinical examination and questioning, data analysis and publication of associated results was obtained beforehand from all children and their legal guardians. 

## 3. Results

### 3.1. There Was No Gender and Age Discrimination among the Participants 

A total of 60 children were included in this study (30 children with FAS and 30 controls). There was no significant difference in gender distribution between the groups (*p* = 0.604). The average age was 8.8 years (SD 1.4) for the FAS group and 8.2 years (SD 1.8) for the controls. There was no significant difference in age between the groups (*p* = 0.122) (Table 1).

### 3.2. No Significant Difference for Exfoliation of Teeth, Swallowing Pattern and Otitis Media 

No significant differences could be found for the exfoliation of teeth with p = 1.000, for the swallowing pattern with *p* = 0.301 (Figure 2a) and for otitis media with *p* = 0.170 (Figure 2b) when comparing children with FAS to the control group. The number of deciduous (*p* = 0.166) and permanent teeth (*p* = 0.309) (Figure 2c, Figure 2d) also was not significantly different between the groups (Figure 2a–d).

### 3.3. Mouthbreathing and Age at Termination of Habits was Significantly Higher for Children with FAS

Significant differences could be found for mouthbreathing with *p* = 0.007, showing a higher number of children with FAS being diagnosed with mouthbreathing (Figure 3a). A significantly higher number of children with FAS had oral habits with *p* = 0.047 (Figure 3b). The age at termination of habits was significantly higher for children with FAS (FAS group: 4.1 years, SD 1.9 versus control group: 2.8 years, SD 1.1, *p* = 0.009) (Figure 3c).

### 3.4. Speech Treatment, Ergotherapy, and Physiotherapy was Significantly More Frequent in Patients with FAS 

Speech treatment, ergotherapy, and physiotherapy were significantly more frequent in patients with FAS with *p* = 0.002 for speech treatment and *p* ≤ 0.001 for ergotherapy and physiotherapy (Figure 3d–f).

### 3.5. A Significant Difference Could be Found for Breastfeeding 

None of the FAS children were breastfed, which was significantly different for the control group (n = 24 with breastfeeding, n = 6 no breastfeeding) with a mean duration of breastfeeding of 6.1 months (SD 5.1). 

### 3.6. Modified DDE Index and DMFT Index were Significantly Different in Patients with FAS 

The modified DDE index was significantly different between the groups (*p* = 0.001) with a mean of 9.5 (SD 9.8) for the children with FAS and a mean of 2.7 (SD 3.6) for the controls (Figure 4b). The DMFT index showed a significant difference (*p* = 0.001) between the groups with a mean of 2.8 (SD 2.4) for the children with FAS and a mean of 0.3 (SD 0.6) for the controls (Figure 4c).

The ratio of teeth with enamel defect and permanent teeth was significantly different between the groups with *p* = 0.001 and a higher ratio for the children with FAS (Figure 4a).

For the ratio of modified DDE with the permanent teeth with enamel defect, a significant difference of *p* = 0.005 could be found with higher ratios for the children with FAS (Figure 4b).

The same applies to the ratio of DMFT with the sum of all teeth showing a significant difference of *p* ≤ 0.001 with a higher ratio for the children with FAS (Figure 4c). 

## 4. Discussion

### 4.1. General Findings 

General dentistry, as well as early orthodontic screening and treatment, seem to be important for patients with FAS, since they showed significantly more problems regarding DMFT, modified DDE, mouthbreathing, and habits when compared with healthy controls. In addition, speech treatment, ergotherapy, and physiotherapy are more frequently necessary in patients with FAS.

### 4.2. Significant Differences in Modified DDE Index and DMFT Index May Be Caused by Abnormal Eating Behaviour in Children with FAS

Concerning the teeth, Naidoo et al. found higher, but no significant results for DMFT in children with FAS [34]. This non-significant tendency could be verified by our study showing significant differences in DMFT for children with FAS.

A study by Amos-Kroohs et al. found abnormal eating behavior in patients with FAS with reduced satiety, constant snacking and more meals per day in comparison with healthy controls [35]. Considering that snacking and in-between meals promote caries decay, this abnormal eating behavior of patients with FAS could be a reason for the higher DMFT index found in our study. In this case, nutritional counseling and further research concerning nutritional habits in patients with FAS would be an advisable contribution to the patient’s (oral) health. 

Animal studies could show that alcohol exposure during pregnancy can influence enamel formation [19,20]. This could be confirmed by our study with a higher modified DDE for patients with FAS.

### 4.3. Reduced Motor Skills Might be a Reason for the Higher DMFT Index in Patients with FAS 

A study by Duval-White et al. identified impairments in handwriting skills in schoolchildren with FAS [36]. This could further explain the higher DMFT levels since toothbrushing skills could be lower in FAS children as well. Another study by Lucas et al. found significant motor impairment in 10% of children with FASD [37,38]. This supports the higher prevalence of physio- and ergotherapeutic treatment found in our study. Reduced motor skills can result in poor oral hygiene which underlines the importance of early physio- and ergotherapeutic measures [39]. 

### 4.4. Higher Need for Speech Treatment May be Associated with Hearing Disorders and Otitis Media in Patients with FAS

Children with FAS have a higher prevalence of hearing disorders, which could be one explanation for the higher need for speech treatment found in patients with FAS in our study [40,41]. A meta-analysis by Popova et al. found a prevalence of 76.2% for expressive language disorders and a prevalence of 81.8% of receptive language disorders in patients with FAS compared with 7.4% in a general US population [42]. This supports the findings in our study that patients with FAS have a significantly higher need for speech treatment.

The same meta-analysis found a prevalence of 77.3% of chronic otitis media in patients with FAS compared with <1% of chronic otitis media in a general US population [42]. This could not be confirmed by our study with a non-significant difference for otitis media in patients with FAS as compared to healthy controls (*p* = 0.170).

### 4.5. Mouthbreathing, Reduced Breastfeeding, and Habits Can be Associated with Malocclusion 

An important aspect is the significantly higher prevalence of mouthbreathing, which could be detected in patients with FAS. Mouthbreathing can result in malocclusion with increased or reduced overjet, anterior and posterior crossbite, openbite, and displacement of contact points between the teeth [28]. 

A recent study by Blanck-Lubarsch et al. found a higher prevalence of crossbites and deficiency in the maxillary region in patients with FAS, which could be a result of mouthbreathing or extended sucking habits [14]. 

The FAS patients in the present study showed significantly prolonged sucking habits with the termination of sucking at a later age than the healthy controls. 

None of the patients in our study was breastfed, which might also be a reason for a higher prevalence of sucking habits [22,23,24] and malocclusions [43,44].

### 4.6. Oral Health Prevention Programs, as Well as Early Interdisciplinary Consultation of Specialists, are Important Measures for Children with FAS 

Taking into account that malocclusion in patients with FAS could be identified in earlier studies and that sucking habits and/or mouthbreathing can promote these malocclusions, it seems even more important to identify sucking habits and mouthbreathing at an early stage. Concerning mouthbreathing in patients with FAS, an orthodontist, as well as an otorhinolaryngologist, should be consulted at an early stage in order to prevent resulting malocclusions. 

Furthermore, it is advisable to include patients with FAS in oral health prevention programs at a dental office with a higher number of appointments, frequent professional tooth cleaning, and more intensive training and instruction concerning oral health and tooth brushing methods. In addition, in these patients, the prescription of special fluoride rinses, concentrated fluoride gels or varnishes on a regular basis could prevent tooth decay. It might be necessary to instruct parents or foster care employees as well in order to help children with FAS with their oral hygiene up to higher ages than healthy controls.

## 5. Conclusions

Our study identified FAS associated problems in the orofacial region with a significantly higher prevalence of mouthbreathing and termination of sucking habits at a later age, as well as significantly higher values for DMFT and modified DDE index. Therefore, multidisciplinary treatment seems necessary in children with FAS in order to prevent developmental problems in later life. Oral health care programs with the administration of fluoride sources, as well as more frequent appointments at a dental and orthodontic office, together with professional tooth cleaning, oral health care instructions and nutritional counseling both for the patients as well as for their legal guardians might prevent damage concerning the teeth and oral health. Concerning mouthbreathing, patients should also be examined by an otorhinolaryngologist at an early age.

## Figures and Tables

**Figure 1 ijerph-16-04401-f001:**
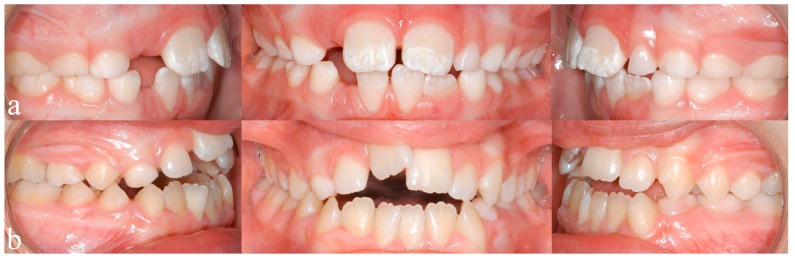
(**a**) Typical enamel opacities, in this case concerning the upper central incisors, (**b**) an anterior open bite configuration resulting from a sucking habit and mouthbreathing.

**Figure 2 ijerph-16-04401-f002:**
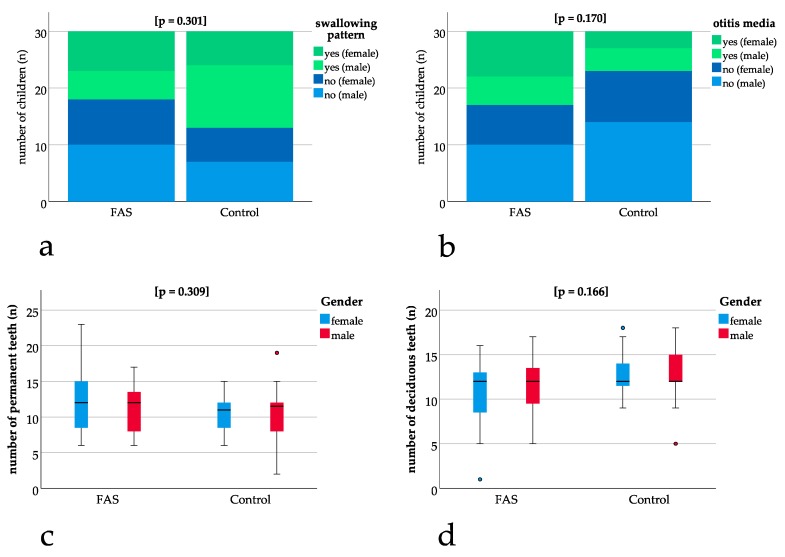
Comparison of children with FAS and controls: (**a**,**b**) bar charts with binary results (yes/no). Positive (“yes”) results are depicted in green (dark green female, light green male) and negative (“no”) results are depicted in blue (dark blue female, light blue male). Bar charts show non-significant differences in swallowing pattern (*p* = 0.301) and otitis media (*p* = 0.170). (**c**,**d**) Box plots for non-significant differences in the number of permanent (*p* = 0.309) and deciduous teeth (*p* = 0.166) thus enabling comparisons concerning dental parameters.

**Figure 3 ijerph-16-04401-f003:**
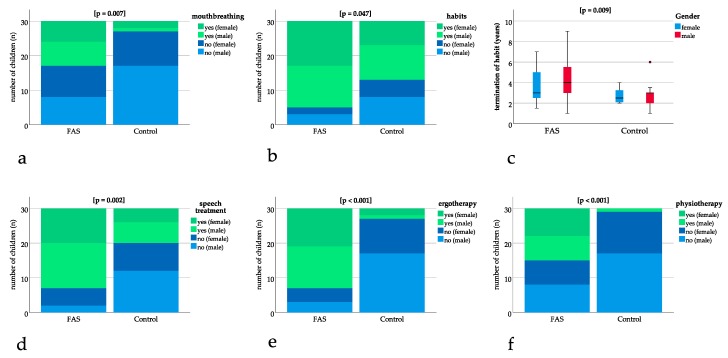
Bar charts with binary results (yes/no). Positive (“yes”) results are depicted in green (dark green female, light green male) and negative (“no”) results are depicted in blue (dark blue female, light blue male). The bar charts show significant differences in (**a**) mouthbreathing (*p* = 0.007), (**b**) oral habits (*p* = 0.047), (**d**) speech treatment (*p* = 0.002), (**e**) ergotherapy (*p* < 0.001), (**f**) physiotherapy (*p* < 0.001). (**c**) Boxplot for significant difference in age at termination of sucking habits (*p* = 0.009).

**Figure 4 ijerph-16-04401-f004:**
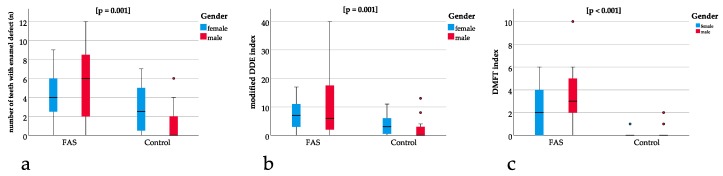
(**a**) Boxplots for significant differences in the number of teeth with enamel defects between the FAS and the control group (*p* = 0.001), (**b**) showing significant differences in modified DDE index between the FAS and the control group (*p* = 0.001). Comparison between (**a**) and (**b**) underline that enamel defects are not only significantly different when measured qualitatively (using the DDE index) but also quantitatively (measuring the absolute number of teeth with enamel defects), (**c**) significant differences in the DMFT index between the FAS and the control group (*p* < 0.001).

**Table 1 ijerph-16-04401-t001:** Descriptive and analytical statistics for all outcome parameters evaluated.

Investigated Parameters	Total	FAS-Group	Control-Group	*p* Value
Gender (n)				0.604 ^2^
Female	27	15	12	
Male	33	15	18	
Age at examination (years)				0.122 ^1^
Mean (SD)	8.5 (1.6)	8.8 (1.4)	8.2 (1.8)	
Median (Range)	8.3 (5.8–11.9)	8.6 (6.6–11.2)	7.6 (5.8–11.9)	
Swallowing pattern (n)				0.301 ^2^
Adult	29	12	17	
Infantile	31	18	13	
Mouthbreathing (n)				0.007 ^2^
Yes	16	13	3	
No	44	17	27	
Oral habits (n)				0.047 ^2^
Yes	42	25	17	
No	18	5	13	
Age at habit termination (years)				0.009 ^1^
Mean (SD)	3.6 (1.8)	4.1 (1.9)	2.8 (1.1)	
Median (Range)	3.0 (1.0–9.0)	4.0 (1.0–9.0)	2.8 (1.0–6.0)	
Speech therapy (n)				0.002 ^2^
Treatment	33	23	10	
No treatment	27	7	20	
Ergotherapy (n)				< 0.001 ^2^
Treatment	26	23	3	
No treatment	34	7	27	
Physiotherapy (n)				< 0.001 ^2^
Treatment	16	15	1	
No treatment	44	15	29	
Breastfeeding (n)				< 0.001 ^2^
Yes	24	0	24	
No	36	30	6	
Duration of breastfeeding (months)				< 0.001 ^1^
Mean (SD)	-	0	6.1 (5.1)	
Median (Range)	-	0	6 (0–24)	
Exfoliation of teeth (n)				1.000 ^2^
Normal	59	29	30	
Dentitio praecox/tarda	1	1	0	
Otitis media (n)				0.170 ^2^
Yes	20	13	7	
No	40	17	23	
Ratio permanent teeth with enamel defect/number of permanent teeth				0.001 ^1^
Mean (SD)	0.28 (0.27)	0.39 (0.27)	0.17 (0.20)	
Median (Range)	0.23 (0–1)	0.36 (0–1)	0.12 (0–0.6)	
Ratio DDE index/number of permanent teeth with enamel defect				0.005 ^1^
Mean (SD)	1.2 (1.2)	1.6 (1.2)	0.9 (1)	
Median (Range)	1.0 (0–4.8)	1.4 (0–4.8)	1 (0–4.3)	
Ratio DMFT index/number of permanent and deciduous teeth				< 0.001 ^1^
Mean (SD)	0.07 (0.1)	0.13 (0.1)	0.01 (0.03)	
Median (Range)	0 (0–0.42)	0.11 (0–0.42)	0 (0–0.1)	
DMFT index				< 0.001 ^1^
Mean (SD)	1.6 (2.2)	2.8 (2.4)	0.3 (0.6)	
Median (Range)	0 (0–10)	2.5 (0–10)	0 (0–2)	
modified DDE index				0.001 ^1^
Mean (SD)	6.1 (8.1)	9.5 (9.8)	2.7 (3.6)	
Median (Range)	3 (0–40)	6.5 (0–40)	1.5 (0–13)	
Number of permanent teeth with enamel defect (n)				0.001 ^1^
Mean (SD)	3.3 (3.2)	4.7 (3.5)	1.9 (2.2)	
Median (Range)	2 (0–12)	4 (0–12)	1 (0–7)	
Number of deciduous teeth (n)				0.166 ^1^
Mean (SD)	11.8 (3.8)	11 (4.1)	12.6 (3.3)	
Median (Range)	12 (1–18)	12 (1–17)	12 (5–18)	
Number of permanent teeth (n)				0.309 ^1^
Mean (SD)	11.3 (3.9)	11.9 (4.2)	10.6 (3.4)	
Median (Range)	12 (2–23)	12 (6–23)	11.5 (2–19)	

^1^ Mann–Whitney U test; ^2^ Fisher’s exact test.

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
