# Peer review of "Tooth Malformations, DMFT Index, Speech Impairment and Oral Habits in Patients with Fetal Alcohol Syndrome"

_ijerph, 2019, doi:10.3390/ijerph16224401_

Round 1

Reviewer 1 Report

The authors have prepared a very informative manuscript on the consequences of FAS. I would recommend acceptance after considering the following minor comment.

1) Non-parametric tests are generally less powerful to detect significance. Please report results of normality tests to justify the use of non-parametric tests.

Author Response

Dear Reviewer,

We would like to thank you for your helpful comments. The manuscript was revised and corrected according to the comments. We highlighted our corrections in the manuscript with yellow colour.

Reviewer 1:

Non-parametric tests are generally less powerful to detect significance. Please report results of normality tests to justify the use of non-parametric tests.

Answer: Thank you for your comment. The pre-analysis of the data shows that:

The null-hypothesis of normally distributed data could be rejected by Kolmogorov-Smirnov tests in most of the subgroups. The skewness of the data is extreme in most of the analysed subgroups

To apply a consistent analysis strategy throughout the dataset we chose a non-parametric analysis strategy for uniform comparisons in all cases.

For clarification, we added this explanation to the statistical analysis section in the manuscript. Please see yellow highlights.

Reviewer 2 Report

The manuscript entitled as “Tooth malformations, DMFT-Index, speech impairment and oral habits in patients with fetal alcohol syndrome (FAS)” by Moritz Blanck-Lubarsch et al., has identified FAS associated dental anomalies or habits among a group of FAS-children. They suggested that children with FAS need early developmental promotion such as speech treatment, ergotherapy and physiotherapy. The authors have managed this manuscript very well. The way of writing is good but there are some major issues need to be addressed. Some of the issues are –

In the abstract, author used several abbreviations like FASD, DMFT, DDE, FAS. It is a general practice not to use any short form in abstract section. Would you please give attention here? In the Introduction section author wrote one/two sentences which represent separate paragraphs. It looks very odd. The author needs to reduce the number of paragraphs in this section. In Figure legends author wrote ‘Figure 1’ but in the text it was mentioned as ‘Fig. 1’. I have noticed this mismatching everywhere in this manuscript. Need to follow a unified pattern that this journal wants. The author needs to recheck all abbreviations they have used. It is recommended to give the full form of any abbreviation at first appearance. In the result section, author wrote ‘Study participants’ as a sub-heading of results. Please, use different sentence to represent your results. Additionally, they need to provide sub-headings for each major finding. The author presented their data in graphical forms. However, they did not provide any error bars for them. Need to add error bars for every bar graph in all figures. In page 5, line 170, the author wrote “Significant differences could be found for mouth breathing with p=0.007, showing a higher number of children with FAS being diagnosed with mouth breathing (Fig. 3a).” However, there is no sign of significance in the Figure 3a. All the bar graphs are lack of any sign of significance. Author needs to provide clear presentation. In the figure legend of Figure (page 5, line 168) author wrote that “2Fig. 2c and 2d showing box plots for non-significant differences in the number of permanent and deciduous teeth.” But in the box plot the author used a steric sign (*). What does it mean? The author also used some dots of different colors but did not mention it in the legend. Author should follow this comment for every figure in this manuscript. Furthermore, the figure legends are very short for every Figure. Author needs to be more informative for each figure. In the Table 1, author wrote ‘Gender, n’; ‘Age at examination, years’ and so on. Did you want to say Gender (n) or Age at examination (years)? Need corrections all over this Table. Furthermore, what is ‘C-Group’? If it is the control group then write it as it is. In discussion section, author needs to discuss their findings with sub-headings and more organized way. The author has used some very old references. Replacement of them to recent one would be appreciated, If possible.

Author Response

Dear Reviewer,

we would like to thank you for your very detailed and helpful comments. The manuscript was revised and corrected according to the comments. We highlighted our corrections in the manuscript with green colour.

Reviewer 2:

In the abstract, author used several abbreviations like FASD, DMFT, DDE, FAS. It is a general practice not to use any short form in abstract section. Would you please give attention here?

Answer: Thank you for your comment. We included the full name of the abbreviations at first mention in the abstract and used the complete term for FASD and FAS in all cases. For DDE and DMFT, the full term is never used in dental terminology which is why we included the full name at first mention for clarity and then used the abbreviation further on. We hope this is suitable, otherwise the word count would also limit the abstract very much (we had to shorten it already).

In the Introduction section author wrote one/two sentences which represent separate paragraphs. It looks very odd. The author needs to reduce the number of paragraphs in this section.

Answer: Thank you for your suggestion. We reduced the number of paragraphs (please see green highlights in the manuscript).

In Figure legends author wrote ‘Figure 1’ but in the text it was mentioned as ‘Fig. 1’. I have noticed this mismatching everywhere in this manuscript. Need to follow a unified pattern that this journal wants.

Answer: Thank you for your comment. For clarity, we changed all abbreviations into the full term.

The author needs to recheck all abbreviations they have used. It is recommended to give the full form of any abbreviation at first appearance.

Answer: We screened the manuscript for abbreviations and added the full form where needed. Please see green highlights.

In the result section, author wrote ‘Study participants’ as a sub-heading of results. Please, use different sentence to represent your results. Additionally, they need to provide sub-headings for each major finding.

Answer: Thank you for your helpful comment. We included additional sub-headings for each major finding according to your suggestions.

The author presented their data in graphical forms. However, they did not provide any error bars for them. Need to add error bars for every bar graph in all figures.

Answer: The bars indicate the actual number of cases for binary (yes/no) results. We are very sorry, but it is not possible to specify an error bar in this case.

In page 5, line 170, the author wrote “Significant differences could be found for mouth breathing with p=0.007, showing a higher number of children with FAS being diagnosed with mouth breathing (Fig. 3a).” However, there is no sign of significance in the Figure 3a. All the bar graphs are lack of any sign of significance. Author needs to provide clear presentation.

Answer: Thank you for your comment. We changed the layout of all our figures for clarification. The significant differences can now be observed more easily.

In the figure legend of Figure (page 5, line 168) author wrote that “2Fig. 2c and 2d showing box plots for non-significant differences in the number of permanent and deciduous teeth.” But in the box plot the author used a steric sign (*). What does it mean?

Answer: Thank you for your question. The steric sign is used for extreme outliers which represent cases that have values more than three times the height of the boxes. For more clarity, we simplified this presentation into dots only since the extreme values do not have any consequences in the respective cases.

The author also used some dots of different colors but did not mention it in the legend. Author should follow this comment for every figure in this manuscript. Furthermore, the figure legends are very short for every Figure. Author needs to be more informative for each figure.

Answer: The dots represent outliers which do not fall into the inner fences of the t-bars. We included this in the statistical analysis section and extended the descriptions of the figures in the figure legends. Please see green highlights.

In the Table 1, author wrote ‘Gender, n’; ‘Age at examination, years’ and so on. Did you want to say Gender (n) or Age at examination (years)? Need corrections all over this Table. Furthermore, what is ‘C-Group’? If it is the control group then write it as it is.

Answer: Thank you for your suggestion. We corrected the table according to your suggestions.

In discussion section, author needs to discuss their findings with sub-headings and more organized way.

Answer: Thank you for your comment. We included sub-headings for clarity, reorganized the structure and added further information to the discussion.

The author has used some very old references. Replacement of them to recent one would be appreciated, If possible.

Answer: We tried to find original references for the indices which is why some very old references were included. We added some more recent references.

Reviewer 3 Report

I found the paper interesting but in need of definitions of dental terms that your broad audience would not necessarily understand.

Author Response

Dear Reviewer,

We would like to thank you for your helpful comments. The manuscript was revised and corrected according to the comments. We highlighted our corrections in the manuscript with turquoise colour.

Reviewer 3:

I found the paper interesting but in need of definitions of dental terms that your broad audience would not necessarily understand.

Answer: Thank you for your comment. For clarity, we added definitions for dental terms as you suggested. Please see turquoise highlights.

Round 2

Reviewer 2 Report

The authors have revised this manuscript very well. They followed all my suggestions as required. Even though, there are some minor points need to be addressed clearly before publication. 

1. Would you please change the title as follows?

"Tooth malformations, DMFT index, speech impairment and oral habits in patients with fetal alcohol syndrome".

2. In the result section, author added some sub-headings which is appreciable. However, I suggest to change the titles of those sub-headings.

For example, they wrote 'Study participants'. But they can switch it to 'There were no gender and age discrimination among the participants'. 

The author wrote 'Mouthbreathing and age at termination of habits'. But they can switch it to 'The Mouthbreathing and age at termination of habits was significantly higher for children with FAS'.

What I want to say is that there should be some differences between the title of sub-headings between 'Material and Methods' and 'Results' sections. Sub-headings of the Result section will represent best finding of interest. I hope that you understand my point. Need to rewrite all sub-headings in the Result section and also follow the suggestion for 'Discussion' section.

Author Response

Dear Reviewer,

we would like to thank you for your positive response and your very detailed and helpful comments. The manuscript was revised and corrected according to the comments. We highlighted our corrections in the manuscript with dark blue colour.

Reviewer 2 :

Would you please change the title as follows?

“Tooth malformations, DMFT index, speech impairment and oral habits in patients with fetal alcohol syndrome”.

Answer: We changed the title according to your suggestion.

2. In the results section, author added some sub-headings which is appreciable. However, I suggest to change the titles of those sub-headings.

For example, they wrote “Study participants”. But they can switch it to “There were no gender and age discrimination among the participants”.

The author wrote “Mouthbreathing and age at termination of habits”. But they can switch it to “The Mouthbreathing and age at termination of habits was significantly higher for children with FAS”.

What I want to say is that there should be some differences between the title of sub-headings between “Material and Methods” and “Results” sections. Sub-headings of the Result section will represent best finding of interest. I hope that you understand my point. Need to rewrite all subheadings in the Result section and also fullow the suggestion for “Discussion” section.

Answer: Thank you for your recommendation. We changed all sub-headings in the results and discussion section according to your suggestion. Please see green highlights in the manuscript.